# Whether Carbon Nanotubes Are Capable, Promising, and Safe for Their Application in Nervous System Regeneration. Some Critical Remarks and Research Strategies

Andrzej Zieliński *[ID] and Beata Majkowska-Marzec [ID]

Department of Biomaterials Technology, Faculty of Mechanical Engineering and Ship Technology,
Institute of Manufacturing and Materials Technology, Gdańsk University of Technology, Narutowicza 11/12,
80-233 Gdańsk, Poland
* Correspondence: andrzej.zielinski@pg.edu.pl

**Abstract:** Carbon nanotubes are applied in or considered for different fields of medicine. Among them is the regeneration or rebuilding of nervous system components, which still lack substantial progress; this field is supported by carbon nanotubes to a great extent as the principal material. The limited research on this issue has involved PU/silk/MWCNTs, PCL/silk/MWCNTs, PCL/PGS/CNTs, chitin/CNTs, PGF/CNTs, CNTs/PGFs/PLDLA, MWCNTs/chitosan, MWCNTs/PPy, PLA/MWCNTs, PU/PAA/MWCNts, GelMA/SACNTs, and CNTs alone, which have been subjected to different surface modifications and applied in the form of solid materials or scaffolds that are degradable or nondegradable. So far, these attempts have shown that the use of surface-modified MWCNTs is a promising way to improve the functions of nervous systems as a whole, even though some drawbacks, such as the potential cytotoxicity or the weak adhesion of CNTs to other components, may appear and be eliminated by their proper functionalization. The present review presents an idea of a nonbiodegradable scaffold structure composed of a chosen conductive polymer that is able to create a scaffold structure, a selected nanocarbon form (with MWCNTs as the first candidate), and a corrosion-resistant metal as a conductor. Other substances are also considered for their ability to increase the mechanical strength and adhesion of CNTs and their biological and electrical properties. The novelty of this approach is in the simultaneous use of nanocarbon and conductive metallic fibers in a polymer scaffold structure.

**Keywords:** nervous system failure and regeneration; carbon nanotubes and fibers; biodegradable and conductive polymers; nanometals; bioactive reinforcing ceramics

## 1. Failures of the Nervous System

The damages to the nervous system belong to the toughest failures of the human body, which frequently become irreversible. Some of them undergo difficult and risky surgeries, and others can only be the object of long-term rehabilitation, but many such accidents cannot be successfully treated. The reasons are due to the high entanglement of the nervous system [1]. That following, the repairing of harmed nerves and the recovery of their functions is still uncertain [2]. Neurological injuries appear to be the most recurrent and need rapid and effective interventions [3].

The nervous system has two components: (i) the peripheral nervous system (PNS) and (ii) the central nervous system (CNS). The PNS includes the cranial, spinal, and peripheral nerves and neuromuscular junctions, and the CNS is located in the brain and spinal cord. The PNS is more prone to impairment because it is not protected by, e.g., hard bone tissue or the blood-brain barrier [4]. On the other side, the recovery of the CNS is slower and less effective than for the PNS due to the deficit of the Schwann cells, which support the development, performance, and regeneration of nerves [5,6]. The mutual contact between bone and nerves is crucial for the correction of large bone defects as well as for the recon-

struction of peripheral nerves using the materials most suitable to ensure osteoconduction, osteoinduction, neuroconduction, neuroinduction, and neuroattraction [7].

Conventional surgery with the use of autografts, allografts, and xenografts is not an excellent medical solution for many reasons. In particular, for autografts, the implanted nerves come from other parts of the body, and it is often difficult to obtain the number of nerves necessary for a full recovery [6]. Moreover, allografts and xenografts create the risk of transmitting diseases that might be followed an implant rejection.

The current research strategies for the development of nerve implants focus on several problems, which include the physical form and structure and the composition of the used implant, its proper application, regeneration efficacy for the nervous system, and the anticipated toxicity of this to different organs. This paper shows some of the selected aspects of the strategy problems proposed based on the state-of-the-art in this field and several of our assumptions, with the most plausible approach for future research, in our opinion, keeping in mind that all these remarks and considerations are highly subjective.

## 2. Biomaterials for the Nervous Systems

As was suggested over 20 years ago, the tissue-engineered nerve construct should have four components: a scaffold construction for the proliferation of axons, Schwann or other support cells, growth factors, and an extracellular matrix [6]. According to another concept, such a conduit must be dissolvable in a biological environment, fast vascularized, and possess low antigenicity, porosity, and be resistant to long-term compressive stress [8]. An engineered "nerve guidance channel" (NGC) that is able to guide the regeneration of damaged nerves was also proposed [9]. Such an NGC is imagined as a tubular device with a single lumen creating the bridge for the reconstruction of the damaged peripheral nerves, which separates the regenerating axons and scar tissue and prevents compression by the surrounding tissue.

For implants, several material classes, like metals and alloys, solid and resorbable ceramics, bioglasses, polymers, carbon (and its organic derivatives), and many composite materials have been proposed [10], yet the nervous system needs specific materials. New synthetic and natural polymers are used most frequently to assemble the perfect scaffold enriched with some cells and growth factors, which can exactly mimic the extracellular matrix (ECM). Such a combination of biochemical, topographical, and electrical signals via various polymers, cells, and growth factors is considered the best solution for efficient regeneration [11,12]. The regarded strategies propose to develop a scaffold similar to the natural ECM that can provide an ideal environment for 3D cell cultures and is fully biocompatible and biodegradable [13]. However, the FDA approved until 2016 [3] only polyglycolic acid (PGA), PCL polycaprolactone, collagen, and polyvinyl alcohol (PVA). Other polymers have been investigated for neurosurgery aims and they include natural compounds such as hyaluronic acid (HA), the benzyl ester of hyaluronic acid (HYAFF), and synthetic polymers such as PLLA (polylactide of lactic acid), PLGA (poly(lactic-co-glycolic acid), polypyrrole (PPy), chitin, and chitosan, polyanhydride, polyhydroxyalkanoate, poly (propylene fumarate) and polyvinyl alcohol [6,9,13,14], (poly(3,4-ethylene dioxythiophene) (PEDOT) [15], and some others which are still under study. Such strategies look reasonable if the nerve conduit implant is designed to undergo a complete transformation. As polymers have several drawbacks, polymer-based composite materials have also been proposed for nerve tissue engineering, including recently bicomponent 3D scaffolds consisting of collagen and chondroitin sulfate [16] and antibacterial cellulose-based composites [17].

Other than only polymeric, materials are increasingly investigated for these considered branches of medicine, with nanotechnology and nano-based materials permanently important. Among them, diverse nanocarbon forms, random and aligned nanofibers, nanopatterning, and inorganic, organic, metallic, and biologically derived nanoparticles of the surface are the most preferred [18]. A need for new materials is especially distinct for long gaps in the nervous conduits [19]. In the past, various strategies have been proposed to provide a bridge between two detached nerve stumps and facilitate the regeneration

of axons [20]. Three generations of such materials include [1] (i) nondegradable materials like silicone conduit, an implant that acts as a tube and connection bridge but must be removed from the body; (ii) dissolvable conduits, like PLA, PVA, and PLGA-based conduits, which, however, act only as a tube and cannot promote nerve regeneration [21–24]; and (iii) conductive materials to form synthetic nerve conduits [25]. Conductive substrates can support cellular activity with or without electrical stimulation [26–28]. The intrinsically conductive and electroactive polymers with easy-to-tailor properties, such as polyaniline (PANI), PPy, polythiophene, polyphenylene sulfide, and oligomers for neural stimulation are among the most popular [29,30]. In addition, nanoparticles, such as gold nanoclusters, graphene nanosheets, and the here-described carbon nanotubes, were applied to increase the conductivity of polymer networks to the desired level of about $10^{-3}$ S/m [31,32].

Despite the materials used, electrical stimulation at various frequencies was shown to enhance the progress of axonal outgrowth in nerves. Moreover, peripheral nerve regeneration in extended gaps occurred effectively after electrical stimulation [33–35], but also pulsed magnetic field and ultrasound waves were utilized [36,37]. Among the different compounds called piezoelectric nano-biomaterials are barium titanate nanoparticles, boron nitride nanotubes, zinc oxide, polyvinylidene fluoride, and fluoride-trifluoroethylene, PLA, collagen, silk fibroin, and graphene oxide [38]. The electrospun conductive PVA)/PEDOT (poly(3,4-ethylene dioxythiophene) scaffolds seem to be one of the best for such a purpose among the composite materials [39].

The electrical stimulation occurs via microelectrodes. When considering the non-carbon materials used for coating, such electrodes are also used as polymers (such as PEDOT [40]) and as composites for nano bioglass/gelatin scaffolds implemented with antibacterial nanosilver and developed as a conduit for peripheral nerve regeneration [41]. The wide application of CNTs for biosensors (for glucose detection) and neurosensors is well-known [42], and they can also be used as biocatalysts, ion channel blockers, tools in cancer diagnosis and therapy, and nanovectors [43]. The electrochemical neurotransmitter sensors capable of detecting dopamine, serotonin, acetylcholine, glutamate, nitric oxide, and adenosine are widely based on carbon-nanostructure-modified electrodes, including carbon nanotubes, graphene and graphene oxide (GO), graphdiyne, and carbon nanofibers (CNFs) [44]. For example, CNTs were applied for flexible neuronal microelectrode devices [45–48], which were important because of their high surface-to-volume ratio. The carbon fiber microelectrodes were covered with a high-surface diamond film to improve neural stimulation [49]. The GO-PANI nanocomposite was shown to be promising as a coating of microelectrodes [50].

The biomaterials for the regeneration of the nervous system can be classified differently. They can be divided [2] into isotropic hydrogel fillers, which provide interluminal support for nerve regeneration; fibrous interluminal fillers, which offer intraluminal topographical guidance for neurites; and patterned interluminal scaffolds, which provide nerve growth with three dimensional (3D) structural support. On the other side, to talk about such biomaterials, it is plausible to consider them in their simplest form, based on either a physical shape or chemical composition, depending on a specific application. However, physical form, composition, and medical destination are interrelated so that the authors have decided (keeping in mind elementary carbon as an essential element of all the here-considered technical solutions) to divide all materials into nonporous (solid) materials and scaffolds, both either prone or resistant to biological degradation. As the title alludes to, we have focused on nanocarbon applications, mainly, but not exclusively, in the form of nanotubes. This is because, despite numerous investigations and proposals focusing on many biomaterials for repairing nervous systems, with a lot of reservations against carbon nanotubes, the last elementary nanocarbon form is still being investigated with growing interest. Therefore, this paper is exclusively limited to biomaterials and mainly to carbon nanotubes (CNTs) destined for the regeneration of the nervous system. Our aim is not only to demonstrate the state-of-the-art in this specific area but to propose a novel strategy for

the future research and development of CNT-based nervous implants, together with the existing hopes, justifications, and apprehensions.

### 3. Carbon Nanotubes

#### 3.1. Forms and Characteristics of Elementary Bionanocarbon

Elementary carbon nanostructures are diverse. The three most popular ones used for the anticipated medical applications are single-walled carbon nanotubes (SWCNTs), multiwalled carbon nanotubes (MWCNTs), and carbon nanofibers (CNFs) [51–59]. Carbon nanotubes were created and described for the first time by Iijima in his well-known article [60]. They are intensively investigated and used for different purposes in medicine and engineering fields because of their significant mechanical, thermal, magnetic, optical, electrical, surface, and chemical properties, in particular, a very high Young's modulus and strength, increased electrical and thermal conductivity, and high chemical activity [61].

These unique properties make CNTs promising candidates for drug carriers in cancer treatment, and in regenerative medicine, bone implants, and nerve restoration [62,63]. Hopley et al. [64], in their nice review, have pointed out that the incorporation of CNTs into polymer scaffolds results in, among other things, increased scaffold strength and flexibility, improved biocompatibility, the retardation of cancer cell division, and the enhancement of angiogenesis. What seems to be the most important, tensile properties of the CNTs gives robust mechanics and dynamics, and their electrical and thermal features are plausible for neural, bone, and cardiac implants. On the other side, carbon fibers were also proposed to make organoids for studying brain-associated neurodegenerative diseases [65].

The CNTs are recommended as components of neural implants as they demonstrate outstanding biocompatibility, cell adhesion, viability, growth, and differentiation, and their high conductivity and mechanical behavior seem advantageous for neural tissue engineering [66–68]. Recently, a major focus has been aimed at MWCNTs applications for nerve regeneration [69]. In the first published paper in this field [70], embryonic rat-brain neurons were grown on MWCNTs, demonstrating that nanotubes, after their functionalization (coating with the bioactive molecule of 4-hydroxynonenal), allowed neurons to develop multiple neurites and enhanced their extensive branching. Afterward, a great amount of research confirmed the notable and subtle effects of CNTs which were attributed to starting axon growth, enhanced by electrical stimulation and contact guidance signals [25,71–86]. The CNTs placed in conductive polymers, e.g., polypyrrole (PPy) [87], polyaniline (PAn), poly (o-phenylenediamine) (PoPD), and polythiophene (PTh) [88], demonstrated remarkable electrical, magnetic, and electronic performance. Moreover, CNTs have also been considered as carriers for nano-drug delivery in chronic neurological disorders such as Alzheimer's and Parkinson's diseases [89].

Such applications become possible due to the CNTs having dimensions comparable to a single protein [90] and being chemically stable, biocompatible, having surface topography mimicking neural tissue, and allowing for the intracellular delivery of biomolecules [71]. Most crucially, neurons and neuronal cell lines can grow and differentiate on CNT substrates [70,80–82,91–95]. The fundamental purpose for such behavior is their high electrical conductivity, in particular, that of purified MWCNTs, which can enhance neuronal circuit network functioning [81,91,96,97]. Besides, functionalized MWCNTs do not alter neuronal morphology and viability [98–100].

Various carbon nanostructures, such as CNTs and CNFs, were synthesized and tested specifically for neural tissue engineering applications [68,82,101–105]. The CNTs remain the most popular form of nanocarbon in medicine, even if there are opinions that CNFs possessing similar properties are cheaper and can be more easily produced [106,107]. The chemical-vapor deposition (CVD) at an elevated temperature is a method commonly used to produce CNTs [95,104], despite some concerns related to the potentially allergic effect of nickel used in the technological process. More advanced free-standing nanostructured matrices were produced by another technology composed of conventional lithography and a layer-by-layer technique [101]. These modified SWCNT structures held up the

outgrowth of neuritis, cell-to-cell communication, and cell differentiation [8]. Patterned microfabricated substrates composed of CNT islands [108] appeared to be a perfectly organized neural network, in which the cells aggregated to the islands of the CNTs, and neuronal cells between neighboring islands grew by connecting axons. The resultant scaffold showed a cohesive structure and excellent mechanical flexibility. The graphitic structure generated by annealing rendered the carbon nanofibrous scaffold with superior electrical conductivity [35]. Fullerenes have also been considered for such a purpose [109].

Another physical form of CNTs, except for fibrous networks, is a mesh or a sponge composed of CNTs [110], able to integrate neuronal networks. For example, in a sponge-like structure, the 3D mesh was made from CNTs, creating the scaffold so as to efficiently regenerate the damaged neural tissue [111]. The carbon nanotubes guided the formation of the nerve fibers, creating a hybrid structure; without a scaffold form, nerve fibers regrew in all directions, weakly creating the bridge between the damaged sections.

CNT-based scaffolds can be obtained by different methods. Scaffolds made by electro-spinning, either by the "sandwich" or dual deposition methods, were highly electrically conductive and cytocompatible and were proposed for several neural applications, such as spinal cord and peripheral nerve regeneration, and even as microfluidic models of the brain [35,112]. It is noteworthy that even if CNTs can be implanted to guide nerve regrowth, these conduits cannot help with the repair of long defects [113]. However, the opposite opinion [114] has also been presented, meaning further research seems necessary as it is a crucial point in neural system regeneration.

As previously mentioned, besides being in the nanoscale range, the most attractive feature of these materials relies on their ability to display metallic and superconducting electron transport properties. However, original CNTs do not have the necessary solubility for their direct application in Medicine. Therefore, it is obligatory to functionalize CNTs, not only to make them more soluble but to allow for their integration into many organic, inorganic, and biological systems and applications without any cytotoxic effects.

### 3.2. Functionalization of Carbon Nanotubes

Two of the main strategies applied to CNTs in physiological conditions include either covalent or noncovalent functionalization [115]. Besides, functionalization, such as topo-logical patterning and electrical stimulation, can yield a significant improvement in nerve guidance conduits [97]. The functionalization of CNTs is important for the prevention of toxicity, and MWCNTs seem safer than SWCNTs [116].

Covalent surface modification can reduce the possible toxicity of CNTs. The noncova-lent functionalization destroys the van der Waals bonds and prevents the aggregation of CNTs into bundles, improving water miscibility [51]. The chemical modification of CNTs changes their surface charge, improving the growth of neurons [89,117–119]. Specifically, the surface charges of MWCNTs can influence the length of neuritis, branching, and the number of growth cones, and the positively charged MWCNTs, in comparison to the negatively charged version, have a higher number of growth cones and neurite branches that promote neurite outgrowth [120–122]. A positively charged surface enhances the electrostatic interactions between the negatively charged plasma membranes of neural cells. Besides, amine groups promote the growth of neurons. In an integrated SWCNT–neuron system, electrical stimulation delivered by SWCNTs can induce neuronal signaling [94]. The growth of the neuronal circuit on a CNT substrate is accompanied by an increment in the network activity related to the electrical conductivity of nanotubes, providing a route for direct electric current transfer and distributing the charge along the surface, which results in a strengthening of the direct electrical bonding between neurons [81,123]. CNT concentration has a critical impact on neurite outgrowth and extension [79]. The incorporation of MWCNTs can result in abnormal neuronal growth and it is important when making surface modification to keep balanced cellular activity [124]. CNTs have a significant affinity for DNA and RNA, creating a complex with the polynucleotides that can be suitable for delivering genes into cells, and CNTs functionalized with PEG improve

blood circulation, reduce retention via the reticuloendothelial system, and stop the binding of proteins [125]. To achieve this goal, CNTs in scaffolds are conjugated with biologically active compounds or with differently charged molecules such as, e.g., 4-hydroxynonenal (4-HNE), a lipid peroxidation product, type IV collagen, and extracellular matrix proteins. Another approach is chemical functionalization via the covalent attachment of functional groups such as neurotrophin [71], polylysine, polyornithine, poly-m-aminobenzene sulfonic acid (PABS), and ethylenediamine [126]. The CNTs chemically functionalized with polyethyleneimine can enhance neural regeneration, in particular, neurite branching, outgrowth, and the attachment of growth cones [127]. MWCNTs can control and promote neurite outgrowth if they are bonded with neurotrophin [71]. For unmodified nanotubes, the neurons extend to only one or two neurites, and after receiving their coating with a bioactive substance, multiple neurites and extensive branching were observed [70].

### 3.3. Surface Films and Membranes

The search for new electrode materials is crucial for improving the long-term performance of neuroprosthetic devices. CNTs have been applied to the coating of electrode sites as CNT electrode sites and 3D CNT probes [128]. One such attempt [129], which involved electrochemically codeposited PPy/SWCNT films, saw the electrode-neural interface characterized by a substantially high charge, low electrode impedance, and reasonable stability, which resulted in cell adhesion and neurite outgrowth. In another neural application, vertically aligned CNFs were coated with PPy films to use them as electrodes [130], demonstrating the improved biocompatibility and electrical and mechanical properties of such a coating. The combination of a conductive polymer (PEDOT) and MWCNTs deposited on the electrode surface (and doped with dexamethasone) resulted in lower in vitro and in vivo impedance values, less neuronal damage, and a reduced inflammatory response [75]. In other research concerning the recording of electrodes [76], a polyimide-based neural interface electrode was coated with gold, showing extremely low impedance and a significant increase in the signal recording resolution. MWCNTs that were applied as coatings on tungsten and stainless steel wire electrodes improved both the recording and electrical stimulation of neural electrodes [131]. Finally, the glassy carbon electrodes modified with bamboo-like CNTs and dispersed into DNA, were shown to achieve highly sensitive and selective quantification of neurotransmitters [132].

The positioning of nanotubes on substrates has importance [133]. When producing horizontally aligned CNTs, the longest neurites on the CNTs tended to align with their direction, although the average neurite length was similar for both CNTs and glass surfaces. Their flexible mechanical properties depend on the length of the CNTs as well as their distance and diameter [134].

In order to avoid the mechanical failure of polymer/CNT composites, freestanding SWCNTs/polyelectrolyte membranes were prepared using a layer-by-layer technique [135]. The tensile strength of such membranes was said to be close to that of ceramics [82]. Besides, SWCNT/polymer films induced cell attachment and differentiation and controlled neurite outgrowth. Such films both maintained the electrophysiological properties of neurons and stimulated neuronal cells for repairing the injured nerves [136]. In other research [91], a similar film was shown to improve the differentiation of neural stem cells into neurons and aided neurite outgrowth.

Some other approaches and compounds were also reported. The layer-by-layer SWCNTs-poly(ethyleneimine), with the last being a polyelectrolyte, seeded with mouse embryonic neural stem cells successfully differentiated into neurons, astrocytes, and oligodendrocytes with the clear formation of neurites [91]. In other research, a poly(acrylic acid)-grafted CNT thin film, was fabricated exhibiting enhanced neuron differentiation and cell adhesion [137]. The structural polarization-controlled neuronal differentiation of human neuronal cells was developed on the CNT monolayer coating, promoting their selective growth [138]. CNTs in conductive polymers were applied to create biosensors for different (also not medical) applications [89]. Flexible 3D carbon nanotubes were proposed

as a peripheral nerve interface [139]. Carbon multielectrode arrays were the best peripheral nerve interface for neural recording and nerve stimulation [140].

### 3.4. Toxicity of Carbon Nanotubes

Cytotoxicity remains the limiting factor for the use of CNTs in biological systems [58]. All nanomaterials can promote cytotoxicity by mechanisms that depend on the type of nanomaterial. Despite the lower cell penetration of carbon nanomaterials, single CNTs can be detected inside cells, and they demonstrate high cytotoxic and genotoxic effects, presumably because of surface chemistry [141]. The bioactivity and cytotoxicity of CNTs are affected by their diameter, length, and functionalization in vitro and in vivo and also by the fabrication method, e.g., a metal impurity such as iron can induce undesirable effects. The genotoxic effect of CNTs via direct contact with DNA was shown to induce mutations in the DNA. The physicochemical properties could then make CNTs toxic to living organisms or the environment [142,143]. The toxicity manifests itself as membrane damage, DNA damage, the appearance of oxidative stress, and changes in the mitochondrial activity and intracellular metabolic routes. The nonbiodegradable nature of CNTs is then the strongest contradiction against their use in implants. CNTs are considered to have carcinogenicity and cause lung tumors, with the effect attenuated by decreasing tube length [144]. The available data provide initial information on the potential reproductive and developmental toxicity of CNTs. MWCNTs likely inhibit the neuronal differentiation of some cells [18]. An exposure of MWCNTs to DRG cultures disturbs regenerative axonogenesis [145,146].

Inhalation of MWCNTs significantly alters the balance between the sympathetic and parasympathetic nervous systems. Whether such transient alterations in autonomic nervous performance would alter cardiovascular function and raise the risk of cardiovascular events in people with pre-existing cardiovascular conditions warrants further study [147]. The toxicity of carbon can be expressed as cytotoxicity, pulmonary toxicity, genotoxicity, dermotoxicity, cardiovascular toxicity, genotoxicity, carcinogenic toxicity, and liver toxicity [146]. Truly, respiratory toxicity is the main concern when carbon nanomaterials are used [148–150]. Other works report nontoxic effects both in in vivo and in vitro [73,83]. MWCNTs are likely to be a more neural-friendly interface than SWCNTs since they allow for a wider external surface and effective functionalization [97].

Nanoparticles can damage and cause cell death due to their small size and large surface area by different mechanisms: creating forms of reactive oxygen species (ROS), rupturing cell membranes, and causing immune responses and chronic inflammation [64]. Four aspects of CNT toxicity have been proposed: (i) increased CNT contents, either in solution or in polymer scaffolds, create reduced cell growth and increased apoptosis, (ii) smaller CNTs show enhanced cell and protein adherence as they are more likely to interfere with cell membranes, but CNT rods that are longer than macrophages (20 μm) will not be phagocytosed and degraded, (iii) synthesis, postfabrication treatments and functionalization influence CNT purity and toxicity, and (iv) each case of cytotoxicity depend on the cell type.

The Fe impurities strapped inside the CNTs produced by CVD may be partially responsible for neurotoxicity generation as they can reduce cell viability and increase the cytoskeletal disruption of cells, diminishing the ability to form mature neurites [151]. CNTs can be manufactured by different methods (arc discharge, chemical vapor deposition, and laser ablation of graphite, among others), and they can have adverse effects due to several other heavy metal nanoparticles, like Fe, Co, Ni, and Y, being present during their synthesis [144–146].

To summarize, carbon nanomaterials may enhance toxicity for two reasons [142]. First, their small size induces permeability changes in cell membranes, enhancing cellular uptake. Second, a high surface area may evoke great chemical reactivity, leading to toxicological responses (positive or negative). But it seems certain that the proper functionalization of CNTs can prevent all forms of toxicity provided that they are not present in excessive

amounts. However, long-term in vivo studies of this grave problem should appear more often than they do currently.

Figure 1 shows the possible advantages and drawbacks of the application of CNTs in medicine. The positives seem to dominate over the negatives, and the last can be eliminated or at least minimized by proper functionalization.

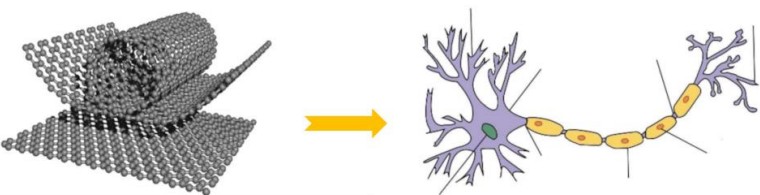

## ADVANTAGES

• Provide a biomimetic nanostructured environment
• Modulate neuronal behavior at either structural (neurite elongation) or functional (synaptic efficacy) level
• Promote guide axons extension, neuron proliferation, differentiation and outgrowth
• Can support sustainable neuronal survival
• Increase network activity of neuronal circuits supporting functional neural circuit growth
• Have synergistic effects on peripheral nerve regeneration
• Stimulate the repair and regeneration of damaged and diseased neural tissues
• Intensively interact with electrically active neuronal tissues

## DRAWBACKS

• May cause asthma and bronchitis, at short exposure and lung cancer at prolongee exposure
• May affect digestive system and gastrointestinal tract enhancing the Crohn's disease and colon cancer
• May cause blood clotting and heart disease in circulatory system
• May be abosrbed by liver and spleen
• The most popular production method may cause allergy because of Ni addition in the process.
• They must be functionalized for an effective application and prevention of cytotoxicity.

**Figure 1.** Scheme of anticipated interaction between carbon nanotubes and the nervous system: advantages and drawbacks. Figures on the top are from: (**left**) Michael Berger. Carbon nanotubes – what they are, how they are made, what they are used for. https://www.nanowerk.com/nanotechnology/introduction/introduction_to_nanotechnology_22.php (accessed on 28 September 2022); (**right**) Alex Bolano. Labeled Neuron Diagram. From: "Neuron" by US National Cancer Institute via WikiCommons CC BY-SA 3.0. https://sciencetrends.com/labeled-neuron-diagram/ (accessed on 28 September 2022).

## 4. CNT-Based Composite Materials

### 4.1. Nonbiodegradable Solid Nanocarbon-Based Composites

There are not many carbon-based composites that can be fully called nonporous and nonbiodegradable. As the first such example [77], an active neural implant was produced by immersing MWCNTs in silicone rubber, followed by etching the surface. Then, the rubber layer was reduced to 13 nm, which covered the CNTs. The fibroblasts and human neuroblastoma cell lines had adequate biocompatibility for the neural implant. In another approach [78], the MWCNTs were incorporated into a polydimethylsiloxane (PDMS) sheet via a printing method to promote the proliferation of the primary neuronal cells. Increased mechanical properties and roughness and superior electroconductivity were observed when compared to PDMS. The new composite material also saw increased adhesion and the proliferation of primary DRG (Dorsal Root Ganglion) cells, and SCs (human peripheral blood mononuclear single cells), referring to poly-l-lysine (PLL), were usually applied to increase cellular attach-

ment. The polymer networks based on polyurethane (PU)/crosslinked polyacrylic acid (PAA) and MWCNTs were also prepared [152]. Considering swelling, mechanical strength, tensile strength, and electrical conductivity, the most plausible content for the CNTs was 1–3 wt. pct. Additionally, the PU/PAA/MWCNT nanocomposites were considered a properly biocompatible material for artificial tendons. Finally, another approach saw the production of a graphene—MWCNTs hybrid material [62]. The hybrid material was not toxic to hESCs (human embryonic stem cells) and could be used for promoting and stimulating the adhesion, proliferation, and differentiation of SCs for neural tissue regenerative treatment. In another study [153], a conductive and nontoxic implant composed of agarose and carbon nanotube fibers allowed for orderly nerve growth, increased differentiation, and the proliferation of neural cells.

*4.2. Nonbiodegradable Carbon-Based Scaffolds*

Such materials are more common than solid neural implants. The reason for this is that scaffolds may make it possible to grow neural cells without any loss of electrical or mechanical properties.

Among the most applied polymers is polycaprolactone (PCL). The scaffolds composed of the MWCNTs and collagen/polycaprolactone (PCL) nanofibers were produced by electrospinning [83]. The composites supported SC adhesion and elongation in vitro. In the in vivo studies, MWCNT-enhanced collagen/PCL conduits were shown to effectively promote the nerve regeneration of sciatic nerve defects and prevent muscle atrophy without body rejection or grave inflammation. Another study [154] focused on aligned PCL/PGS (polyglycerol sebacate) fibers containing different amounts of MWCNTs, fabricated via electrospinning. The Young's modulus, ultimate tensile strength, wettability, and water uptake values increased for the observed scaffolds with rising MWCNTs content. A quite similar approach [105] demonstrated a fibrous composite based on PCL, and either CFs or SWCNTs. The PCL/SWCNTs nanocomposite was especially satisfactory in creating a suitable environment for endothelial cells. Additionally, in [137], PCL/graphene and PCL/CNTs were tested as conductive nerve guidance conduits in sciatic nerve repair. The PCL/graphene/CNTs composite exhibited a higher number of axons and nerve areas when compared to PCL scaffolds and either graphene or CNTs. Furthermore, the addition of graphene to the PCL/CNT composite facilitated the CNT dispersion within the PCL, and enhanced nerve regeneration. Two-dimensional thin film scaffolds composed of biocompatible polymer-grafted carbon nanotubes (CNTs) can selectively differentiate human embryonic stem cells into neuron cells while maintaining excellent cell viability.

Another highly interesting conducting polymer is the polyurethane (PU) used in [155], together with polypyrrole (PPy) for the formation of a self-electrical stimulated double-layered nerve guidance conduit (NGC), assembled from electrospun mats with an aligned oriented inward layer covered with a random oriented outer layer. The biomimetic NGC was achieved from chitosan-grafted PU with well-dispersed functionalized MWCNT nanofibrous mats after a uniform coating of PPy. The structural framework of the NGC exhibited a cellular biomaterial interface improving the electrical conductivity, mechanical strength, and cytocompatibility, in particular, the regrowth, proliferation, and migration of Schwann cells. Nanostructured fibrous scaffolds based on flexible thermoplastic PU and surface-functionalized MWCNTs were also produced by electrospinning [156]. A linear correlation between electrical conductivity and cell signaling, and neural gene expression was found, and an increase in both bulk and surface modulus was due to the presence of MWCNTs in the scaffolds. In similar research [157], fibrous scaffold PU-silk fibroin-functionalized MWCNTs showed the important electrical conductivity and absorption of extracellular matrices (ECM), and in the aligned scaffolds, the substantial growth and proliferation of SCs.

Chitosan is a natural polymer frequently considered for medical applications, even if its properties strongly depend on its deacetylation degree and are hardly

repeatable. Such composite scaffolds [85] were created by aligning MWCNTs in a chitosan scaffold fabricated by an electric-field-alignment technique. Not only were the mechanical properties greatly improved, with the incorporation of only 0.5 wt. pct. of the aligned MWCNTs, but also electrical conductivity increased 100,000 times along its direction. In [158], an MWCNT/chitosan scaffold (highest conductivity) and GNP (graphene platelets)/chitosan scaffold (lowest conductivity) were obtained. The hybrid scaffolds showed increased elastic modulus and ultimate tensile strength over the MWCNT/chitosan and GNP/chitosan scaffolds. Considering the potential cell adhesion, the MWCNT/chitosan composites were more effective than GNP/chitosan and both revealed cytocompatibility.

### 4.3. Biodegradable Solid Nanocarbon-Based Composites

Guidance conduits (to nerve cells) prepared as composites of MWCNTs and poly(2-hydroxyethyl methacrylate) (pHEMA) composites showed effective adhesion, activity, and viability to biological cells [159]. In another work, a composite of silk, fibronectin, and SWCNTs resulted in ideal bioactivity and a subsequent nerve guidance conduit [160]. Three-dimensional collagen–fibrin–MWCNT composite materials were also developed [161]. Investigating the PCL/nanocarbon/graphene composite, a two-fold increase in the number of myelinated axons was found for this prosthesis of the sciatic nerve together with muscle atrophy [162].

### 4.4. Biodegradable Nanocarbon-Based Scaffolds

Some totally or partially biodegradable porous or fibrous structures were used to construct an implant for the nervous system. They include some polyesters, mainly poly(glycolic acid (PGA), PLA, poly(lactic acid–glycolic acid) (PLGA), and only some polyurethanes. Additional nondegradable materials include natural fibrin, collagen, keratin, alginate, chitin and chitosan, and silk fibroin, as well as extracellular matrices [2]. The idea of these technical solutions is to combine two substances: the first, which forms a skeleton, and the second, which may degrade to allow the nerves to grow. Surprisingly, in this specific review, no nanocarbon forms have been mentioned, even if a lot of these composites have already been developed [54].

PLA and PLLA are likely the most often used biodegradable polymers. For the conductive conduit prosthesis (nine weeks after implantation), the tissue-engineered construct made of a rolled sheet of SWCNT/PLLA nanofibrous scaffolds, with some implemented cells, promoted the axonal outgrowth and regeneration of peripheral nerves [163]. A similar nanocomposite scaffold [74] was obtained by dispersing the MWCNTs in a PLA matrix to provide electrical signals and mimic neural topography. A more complex scaffold [164] for a neural guidance conduit was prepared from PLLA and modified MWCNTs and was filled with SCs and nanocurcumin-encapsulated chitosan nanoparticles. Controlled curcumin release decreased SC apoptosis, enhancing the regeneration of injured peripheral nerves. A significant increase in the number of axons in the damaged sciatic nerve and a compelling fall in the number of vessels in the fibrin groups were detected by histological testing. The fabrication of nerve guidance channels was also created using PLA/MWCNTs/gelatin nanofibrils coated with recombinant human erythropoietin-loaded chitosan nanoparticles [165]. In the work carried out by the authors of [84], laminated CNTs were chemically linked onto the surface of aligned phosphate glass microfibers (PGFs). The CNT-interfaced PGFs (CNT–PGFs) were successfully placed into 3D poly(L/D-lactic acid) (PLDLA) porous tubes by wrapping the CNTs-PGFs onto a PLDLA nanofiber mesh embedding them into a porous PLDLA tube afterward.

### 4.5. Carbon Nanotubes on Solid Substrates

CNT networks deposited on solid substrates were also used for the directed growth and differentiation of hMSCs, which could recognize the arrangement of individual CNTs in the network and grow in the direction of the CNT alignment [166]. In other research [28], a CNT rope substrate was developed and seeded with neural stem cells. After electrical stimulation, neurite elongation and increased differentiation of NSCs into neuronal cells appeared.

### 4.6. Hydrogels

The chitin/CNT composite hydrogels [73] demonstrated enhanced tensile strength and elongation and decreased swelling when compared to chitin only. They also exhibited good hemocompatibility, biodegradation in vitro, biocompatibility, and no cytotoxic effect. Besides, the promotion of adhesion in those implants enriched with calcium ions in the form of tubular hydrogels was obtained [72]. Other technical solutions [167] constituting methacrylated gelatin (GelMA) hydrogel and super-aligned carbon nanotube sheets (SACNTs) were used to form a material that showed good biocompatibility (GelMA) and conductivity for CNTs. In another work [79], SWCNTs were applied to control the electrical properties of a collagen-based composite hydrogel. Another hydrogel was produced via a multistage complex technique and was composed of reduced graphene oxide (rGO), CNTs, oligo(poly(ethylene glycol) fumarate) (OPF), and 2-(methacryloyloxy)ethyltrimethylammonium chloride (MTAC) [63]. In [168], a biomimetic core-shell scaffold based on aligned conductive nanofiber yarns (NFYs) within a hydrogel was developed. The aligned NFYs were composed of PCL, silk fibroin (SF), and CNTs. Such a 3D hierarchically aligned core-shell scaffold mimics the nerve fiber structure and positively affects the alignment and extension of neurites, with the hydrogel shell protecting nerve cell organization within a 3D environment/In the other research,, a highly stable and uniform dispersion of multiwall carbon nanotubes in an aqueous solution was applied to prepare [169] CNT/sericin hydrogel, in [170] multiwall carbon nanotube/gelatin–polyvinyl alcohol nanocomposites, with varying MWCNT content using solution casting, and in [171] gelatin-chitosan hydrogel. The desired effect of the addition of the MWCNTs on the mechanical, thermal, and swelling properties of the gelatin-chitosan composites was achieved. The covalent functionalization of MWCNTs facilitated the interfacial interaction between the natural polymer blend and the nanotubes, which further enhanced the dispersion within the matrix and, thus, ultimately enhanced the mechanical properties of the blends. The surface and interface structures of the composites were studied by SEM, and the intimate relationship between the structure and the overall performance of the composite was revealed. The thermal, swelling, and drug-releasing properties were also found to be superior compared with the gelatin-chitosan blend due to the addition of nanotubes. Besides the effectiveness of the drug release rate, the prepared MWCNTs/gelatin-chitosan nanocomposites have not shown any cytotoxicity, and it is believed that such nanocomposites can be employed as targeted drug delivery agents in nanomedicine, targeted thermal tumor ablation, and the magnetic field-targeting of tumors. The directional growth of hMSCs follows, as a rule, the alignment direction of the individual CNTs.

The different materials applied for the regeneration of the nervous system are shown in Figure 2, and their different physical forms are listed in Figure 3.

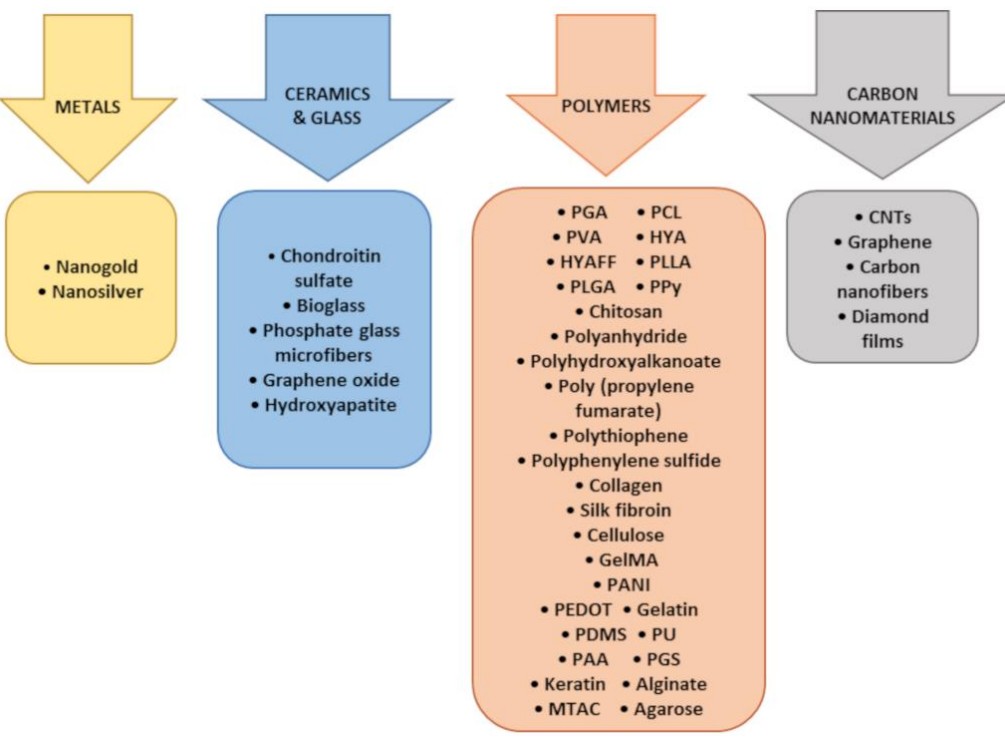

**Figure 2.** Materials used for applications in the regeneration of the nervous system.

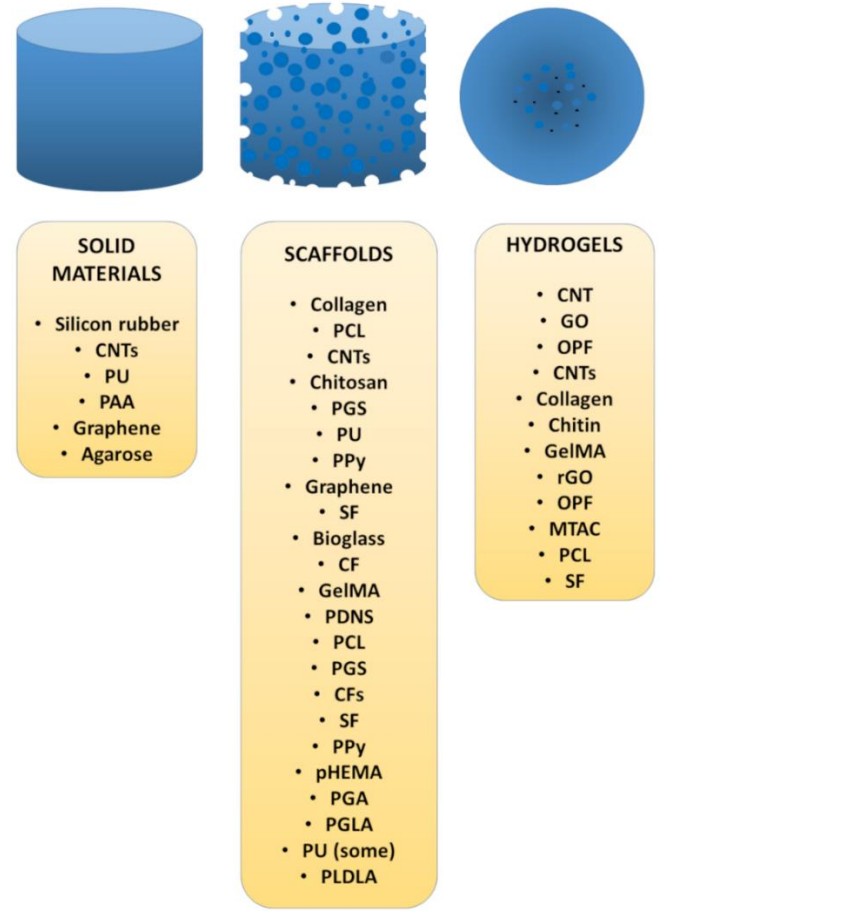

**Figure 3.** Different physical forms and chemical compositions of biomaterials for the regeneration of the nervous system.

## 5. Anticipated Research Strategy for Nervous Conduit Implants

For a start, let us consider what qualitative/quantitative indices should be the most plausible for the specific application of an artificial nervous conduit.

It should have sufficient conductivity, at least 3 S/cm. This condition needs the application of either carbon nanostructures or conductive polymers or some metallic structures. We propose to use the following materials: (i) MWCNTs functionalized with a chosen organic compound and carbon nanofibrils and/or graphene, (ii) polypyrrole (PPy) and/or poly(3,4-ethylenedioxythiophene), and poly(styrenesulfonate) (PEDOT: PSS); among the conducting polymers; (iii) gold, platinum, silver, and/or copper wires, as noble or seminoble metals. So far, another applied solution is the use of biodegradable scaffolds without any conductive components, besides the use of CNTs, whose electrical properties increase after biodegradation, followed by gradual angiogenesis and the growth of neural axons. A similar effect can be expected from stable scaffolds. The question remains whether each patient's body, possessing mechanical failures in the nervous system, will be capable of developing a novel nervous network in a reasonable time and whether a gradual decrease in scaffold mechanical strength will positively affect the delicate implant construction, which might destroy the body. We think that the conductive nervous implant based on permanent polymeric scaffolds is the best solution, at least for aged patients, or, better, for those whose regeneration systems can be insufficient to create the electrical circuits by the growth of new conduits. Therefore, the highly conductive implant seems a good alternative compared to biodegradable scaffolds or stable solid implants.

The implant should demonstrate no cytotoxicity. For all the above-mentioned conductors, and in particular, for CNTs, such fairs (justified or not as the experiments have given different results, and the results obtained in vivo on animals can be the same for human beings, also depending on individual patient's features) appear. Carbon nanotubes are often declared potentially carcinogenic. However, cancer cells appear in living bodies for different reasons and grow abnormally fast only in some conditions. The use of small quantities of CNTs, especially less toxic functionalized MWCNTs, seems reasonable. However, concerning the remaining CNTs, as important mechanical constructs, the electrical conductance can be, according to our strategy, enhanced by carbon nanofibers.

The application of some polymers is reasonable, but it seems reasonable to select only conducting polymers that have already proved their importance in forming scaffolds by electrodeposition or patterning, or even 3D printing, as being the most likely. We are going to look for other conducting polymers which fulfill four conditions: (i) are good conductors, (ii) can form scaffolds, (iii) are nonbiodegradable, and (iv) are nontoxic.

Finally, noble and seminoble metals can be supposedly safely used inside a living body to increase the conductivity of a nervous implant. It is true that all metals, which means over 90% of all elements, present in an elementary form (and not either in inorganic (bone tissue) or organic (blood with chelated iron) compounds) have upper limits, above which they might become toxic. However, in the expected applications, the corrosion dissolution is very low, even after many years. Besides, the proposed amounts of metals are also limited.

Passing over to the recapitulation and conclusions, it seems necessary or at least plausible to show all important quantitative indices from the references here emphasized. Table 1 illustrates this and the considered properties. It can be well-observed that there are still many specific points that have not been characterized yet. On the other hand, some data look discrepant, which can be certainly attributed to substrates and their characteristics, process parameters, applied methods, and several other factors.

**Table 1.** Mechanical and electrical properties of CNT-implemented coatings and scaffolds.

| Components | Morphology | Ultimate Tensile Stress or Compression Strength ** Elastic Modulus at Tension * or Compression ** [MPa] | Electrical Conductivity S/m $\times 10^{-3}$ | Reference |
|---|---|---|---|---|
| Carbon rope with MWCNTs | Elevated and lowered ridgelike structures propagating through the surface of the CNT rope in a spiral direction; the bundles of CNTs of diameters of about 20 nm. | - | - | [28] |
| GO and PANI nanocomposite coatings on titanium | The presence of two phases derived from GO and PANI stacked on top of each other to form a laminate | | | [50] |
| CNT-GO-OPF hydrogel | Four types of hydrogel: neutral transparent, positively charged transparent; opaque with dark color from carbon; positively charged opaque (conductive) | 0.56–58 0.60–0.80 ** | 0.31–5.75 | [63] |
| Chitosan-CNT-HAp hydrogel | Semi-transparent, network morphology | 0.50–0.57 * 0.72–0.78 * | - | [72] |
| CNT-polymer scaffold | Uniform coatings with nanofibrous morphology | - | - | [74] |
| SWCNT-collagen hydrogel | - | - No effect on bulk modulus | | [79] |
| Collagen/PCL/MWCNTs scaffold | Fibrous meshes and porous conduits | 4.5 * - | - | [83] |
| CNT-bioglass scaffold | Functional arrangement: the microfibers packed inside, the thin wrapping sheet, and the slightly thicker outermost layer | - | - | [84] |
| CNT-chitosan scaffold | Uniform black films, macroscale uniformity of aligned CNTs | 50–75 * 1600–1650 * | 0.84–5.25 | [85] |
| PCL/PGS/MWCNT scaffold | Bead-free and uniform aligned fibers | 0.7–1.1 * 0.29–0.41 * | - | [154] |
| MWCNT/PU/PPy/chitosan mat | Random and aligned fibrous mats | 14 * - | - | [155] |
| PU-CNT scaffold | MWCNT particles oriented along the fibers' axis | 13.17–20.57 * 3.94–10.01 * | 9–31.5 | [156] |
| PU-SF-MWCNT scaffold | MWCNT particles dispersed along the fibers' axis | 16 * - | - | [157] |
| MWCNT-graphene-chitosan scaffold | Tubular morphology for MWCNTs, and GNPs appearing as wrinkled nanoplatelets in the chitosan matrix | 80–90 * 2700–3200 * | 0.005–0.019 | [158] |
| MWCNT-pHEMA membrane | The MWCNTs were randomly distributed on the hydrogel surface; some of the nanotubes formed clumps and some were dispersed | 1.25–2.0 * 0.32–0.41 * | - | [159] |
| SF/MWCNT/FN tubular nerve guide conduit | Aligned fibronectin containing nanofibers on freeze-dried silk fibroin/SWCNT substrates | - | 2.1 | [160] |
| CNT-GelMa scaffold | The ordered and parallel arrangement of the super-aligned carbon nanotubes with a diameter of 30–50 nm | - | - | [167] |
| CNT/sericin hydrogel | Interconnected porous microstructure | 0.04–0.07 * 0.034–0.76 * | 0.03–0.39 | [169] |
| CNTs-Gelatin-PVA | MWCNTs are homogeneously distributed into the nanocomposites matrices and the increase in CNT loading progressively blackened the blend nanocomposites. | 90 * 641 * | 0.0085 | [170] |
| CNT-PDMS (poly(dimethylsiloxane) | - | 4.3 * 3.6 * | | [172] |
| Hyaluronan/CNT hydrogel | Porous structure with a mean pore size decreasing in the presence of MWCNTs | 0.017–0.067 ** in the low compression zone and 0.45–0.60 in ** the high compression zone | Conductivity is slightly lower than of hyaluronan alone | [173] |

Table 1 illustrates that mechanical properties are sometimes highly different. The reason is that these material solutions appear as either scaffolds, coatings, membranes, or mats and ropes. Among the coatings, the mechanical properties are similar. For scaffolds, the differences are sometimes great, in particular, the Young's modulus is particularly high (i.e., the material is not deformed even at high loads) for the CNT-chitosan and MWCNT-graphene-chitosan scaffolds.

Table 2 shows the results of some of the biochemical tests. There are several differences and similarities between the tests.

**Table 2.** Biological properties of CNT-implemented coatings and scaffolds.

| Components | WST-1 Test [%] | LDH Test [%] | MTT Test [%] | Live/Dead Assay [%] | Reference |
|---|---|---|---|---|---|
| Carbon rope with MWCNTs | 103–105 | About 100 | | > 90 | [28] |
| GO and PANI nanocomposite coatings on titanium | | The cytotoxic effect after 24 h observed for GO:PANI (1:1) weight ratio | | | [50] |
| CNT-GO-OPF hydrogel | 100 | | | 80–102 | [63] |
| Chitosan-CNT-HAp coating | | | 97 to 112 | | [72] |
| Chitin/CNT hydrogel | | | 100–115 | | [73] |
| CNT-polymer scaffold | | About 100 | | | [74] |
| CNT-bioglass scaffold | 105–110 | | | | [84] |
| CNT-chitosan scaffold | | | 85–145 | | [85] |
| MWCNT-PAA scaffold | About 100 | | | | [137] |
| PCL/PGS/MWCNT scaffold | | | 120–220 | | [154] |
| MWCNT/PU/PPy/chitosan | | | | 140–220 | [155] |
| PU-CNT scaffold | | | 170–200 | | [156] |
| MWCNT-graphene-chitosan scaffold | | | 65–140 | | [158] |
| MWCNT-pHEMA membrane | | | 14–124 | | [159] |
| SF/MWCNT/FN tubular nerve guide conduit | | | 105–110 | | [160] |
| CNT-GelMa scaffold | 95–120 | | | About 100 | [167] |
| CNT/sericin hydrogel | | | | About 100% | [169] |
| CNT-PDMS (poly(dimethylsiloxane) | | | No significant decrease in viability | | [172] |
| Hyaluronan/CNT hydrogel | | | No significant effect | | [173] |

The WST-1 assay comprises the reduction of the tetrazolium salt WST-1 to formazan. It is applied for the measurement of cell mitochondria functionality. The CCK8 test is similar with some small modifications.

The LDH assay measures the value of the enzyme lactate dehydrogenase (LDH), which is released upon cell lysis. It is an index of necrosis.

The MTT assay assesses cell metabolic activity. Some cellular oxidoreductase enzymes reflect the number of viable cells and can reduce the tetrazolium dye MTT to insoluble formazan of a purple color. The assay is also similar to the WST and CCK tests, differing in some chemicals and procedures.

The Live/Dead assay is a cell staining procedure. Live cells are stained with calcein and demonstrate green fluorescence upon the excitation evolved by their cytoplasm. The dead cells are labeled with the ethidium homodimer dye (EthD) bond to their DNA and express the red fluorescence.

As can be seen, in almost every test, the viability of the tested cells has not demonstrated cytotoxic effects. However, the effects of different materials on adhesion, migration, and proliferation of cells have often been positive, yet they are shown here as being hardly comparable to each other.

Summarizing our considerations, the most fruitful and safe material strategy can or should (from our research planning) include MWCNTs and carbon nanofibers, with some

conductive polymers (at least one among those proposed so far and recommended for medical biodevices) and one of the four corrosion-resistant metals here mentioned. The future strategy will focus on the optimal composition of this conduction part of an implant, taking as criteria the conductivity, cytotoxicity, and mutual adhesion.

The shown research strategy defines, to some extent, the discrimination criteria and materials to be investigated. The main criteria in vitro for an assessment of the tested solutions are the following: (i) electrical conductivity, (ii) mechanical properties in long-term tests, (iii) cytotoxicity, and (iv) biocompatibility defined as in vitro wettability. In addition, we plan biological tests in vitro to monitor the growth of neural cells and the number of axons. Besides, for mechanical strength, only carbon nanotubes, carbon nanofibers, and possible graphene among the discussed nanocarbon-based conductive materials can likely be purposeful. On the other hand, all elementary carbon forms have poor adhesion to the rest of the composite, and cracks can be initiated at their interface. Therefore, CNTs are usually (and must be) surface-modified to enhance their adhesion and prevent cytotoxicity. Besides that, we plan to strengthen the composite materials by the addition of some ceramics, in particular, a nano form of reduced graphene oxide (rGO), other nano oxides, or nanohydroxyapatite.

The composite material for nervous conduits should demonstrate antibacterial properties. Such an effect can be obtained by implementing metals, but this hypothesis will be verified.

Finally, each strategy should focus not only on the optimization of material properties but also take into account the material and manufacturing costs. Therefore, each research piece and proposal for new materials will also calculate such costs.

## 6. Future Perspectives

Based on the presented literature review, it can be concluded that carbon nanotubes are a promising material candidate for applications in the regeneration of nerve tissues due to their unique electrical, thermal, and mechanical properties. They are chemically stable and biocompatible, and their topography mimics nervous tissue, which makes them an ideal substrate for neurons and neuronal cell lines. To avoid toxic effects and to better join the CNTs and polymeric scaffold, the MWCNTs subjected to obligatory chemical functionalization should be applied as the first component. Moreover, functionalized CNTs will change neither the morphology nor viability of neurons. The functionalization of CNTs is also important as it affects their surface charge, improving the growth of neurons. Making CNTs positively charged will promote neurite outgrowth and strengthen the direct bonds between neurons.

The anticipated optimized artificial neural conduit should include MWCNTs and carbon nanofibrils to elevate the electric conductance and strengthen the scaffold; one or two have already been recommended for medical biodevices and are not prone to degradation, with noble or seminoble metals used to increase the electric signals. The nerve conduit would be best produced by rolling polymer(s) film with nanocarbon forms and metallic wires set into the strong mechanical tube. The future strategy will focus on the optimal composition of this conduction part of an implant, taking as criteria the conductivity, cytotoxicity, and mutual adhesion, with focus on the development of the fabrication procedure of these small implants.

Various forms of CNTs, such as sponges, 3D meshes, and CNTs islands, might also be considered for making scaffolds for nerve substitutes. Additionally, CNTs are implanted to direct nerve regrowth. The SWCNT/polymer films have also been shown to improve the differentiation of neural stem cells into neurons and aid neurite outgrowth.

The main criteria for the optimization of the tested material solution should include electrical conductivity, mechanical properties (in long-term tests), cytotoxicity, and biocompatibility defined by wettability. Special biological tests should monitor the growth of neural cells in vitro and the number of axons. The other components of the scaffold, such as graphene oxide, an increase in mechanical strength, and the growth factors should also

be considered. These in vivo studies are necessary to accurately verify the achievement of a substantial improvement in nervous conductivity.

Each strategy should also take into account the material and manufacturing costs, which are usually the ignored aspects of research work.

**Author Contributions:** Both authors equally contributed to this paper. In particular: Conceptualization, A.Z. and B.M.-M.; methodology, A.Z. and B.M.-M.; resources, A.Z. and B.M.-M.; writing—original draft preparation, A.Z. and B.M.-M.; writing—review and editing, A.Z. and B.M.-M. All authors have read and agreed to the published version of the manuscript.

**Funding:** This research received no external funding.

**Institutional Review Board Statement:** Not applicable.

**Informed Consent Statement:** Not applicable.

**Data Availability Statement:** Not applicable.

**Conflicts of Interest:** The authors declare no conflict of interest.

## Abbreviations

CNS—central nervous system; CNF(s)—carbon nanofibers; CNT(s): carbon nanotubes; FN: fibronectine; GelMA: Gelatin-methacryloyl; GO: graphene oxide; HA—hyaluronic acid; HAp: hydroxyapatite; hMCS—human mesenchymal stem cells; HYAFF—benzyl ester of hyaluronic acid; MWCNT)(s)—multiwall carbon nanotubes; NGC—nerve guidance channels; OPF—oligo(poly(ethylene glycol) fumarate); PAA—polyacrilic acid; PCL—polycaprolactone; PANI—polyaniline; PCL—polycaprolactone; PEDOT—poly(3,4-ethylene dioxythiophene); PGA—polyglycolic acid; PGS—poly glecerol sebacate; pHEMA—poly(2-hydroxyethyl methacrylate; PCL—polycaprolactone; PLGA—poly(lactic-co-glycolic) acid; PLA—polylactic acid; PLLA—polylactide of lactic acid; PNS—peripheral nervous system; PPy—polypyrrole; PU—polyurethane; PVA—polyvinyl alcohol; PU—polyurethane; PPy—polypyrrrole; pHEMA—poly(2-hydroxyethyl methacrylate; OPF—oligo(poly(ethylene glycol) fumarate); SF—silk fibroin; SWCNT—single-wall carbon nanotubes.

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
