# Peer review of "Whether Carbon Nanotubes Are Capable, Promising, and Safe for Their Application in Nervous System Regeneration. Some Critical Remarks and Research Strategies"

_coatings, doi:10.3390/coatings12111643_

Round 1
Reviewer 1 Report
This paper tries to provide a thorough review on exploring that carbon nanotubes whether or not are applicable in nerve system regeneration. The topic is current and deals with a key issue of trying to find a way to cure the nerve system failures.
However, unfortunately, this manuscript is poorly written, the English grammar is not clear, hard to understand, and this often leads to confusion. It makes an impression as if it was written in a hurry. For example, the title of Section 3 and 4 is the same.
Moreover, the reference is not up-to date, there are only a few recent article cited so it needs to be updated.
Placing a Table of contents is advisable at the beginning of the manuscript (after the abstract)
In general, since the topic is really interesting and current, the paper might be published only after major revision, and fully re-written and have the manuscript checked by native English speaker in order to avoid any misunderstanding or misleads.
Author Response
Reviewer No. 1
- This manuscript is poorly written, the English grammar is not clear, hard to understand, and this often leads to confusion. It makes an impression as if it was written in a hurry. For example, the title of Section 3 and 4 is the same.
Answer: The manuscript has been thoroughly checked again by both of us and with the Grammarly software, and other obvious errors as this indicated by the reviewer has been corrected. There has been no time to look for any native speaker, but several of us speak fluent English and we do not feel that now the manuscript is hard to read and understand. Hopefully, the reviewer will accept the improved manuscript.
- Moreover, the reference is not up-to date, there are only a few recent article cited so it needs to be updated.
Answer: We have already very carefully reviewed the Scientific Direct database. However, there have been not many papers that can be seen as the keywords “carbon nanotubes, implants, nerve system” proposed. Considering this remark, we have reviewed six last years and, apart from Elsevier, also Wiley, Taylor, and Francis, Web of Science, MDPI, and others. There are not many new papers recently published directly in this area which we have not noticed, we presume. However, we have found and cited several papers which are close to the considered subject and support our discussion or introduction clarity. Now we hope that the literature on this relatively limited subject is more recent and could be accepted.
- Placing a Table of contents is advisable at the beginning of the manuscript (after the abstract)
Answer: We have done it.
All changes have been shown in the revised manuscript (version “Detect changes”). Besides, we have put to the resubmitted manuscript also the new version after the recommendation “Accept all changes”.
Reviewer 2 Report
This review summarized the CNTs for improving functions of nervous systems as a whole even if some drawbacks such potential cytotoxicity or weak adhesion to other components. Various CNTs, such as sponges, 3D meshes and CNTs islands, work well as scaffolding for never fibers. The authors proposed a combination of MWCNT and carbon nanofibrils to ensure corrosion resistance and antibacterial properties. Overall, this is a comprehensive review with many latest developments. There are a few suggestions for authors’ evaluation.
1. A new scheme to summarize the relationship between CNTs and nervous system is suggested to be created.
2. More graphical data should be provided in the main text.
3. The morphological data need to be included in Table 1 and Table 2.
4. More parameters on electrical conductivity have to be involved in Tables.
5. The functionality of CNTs on specific nerve function enhancement should be included, such as improvement of NSC differentiation and growth, etc.
6. The current drawbacks should be discussed in Outlook section or Conclusion.
Author Response
Reviewer No. 2
- A new scheme to summarize the relationship between CNTs and the nervous system is suggested to be created.
Answer: We have made it.
- More graphical data should be provided in the main text.
Answer: We have added two more figures.
- The morphological data need to be included in Table 1 and Table 2.
Answer: It has been done.
. More parameters on electrical conductivity have to be involved in Tables.
Answer: It has been done.
- The functionality of CNTs on specific nerve function enhancement should be included, such as improvement of NSC differentiation and growth, etc.
Answer: We have made it.
- The current drawbacks should be discussed in Outlook section or Conclusion.
Answer: It has been done
All changes have been shown in the revised manuscript (version “Detect changes”). Besides, we have put to the resubmitted manuscript also the new version after the recommendation “Accept all changes”.
Reviewer 3 Report
The authors have come up with some really interesting information to write the article. However, as I read the whole thing, I only found words and words everywhere and lost myself in the sea of words a few times. My first recommendation would be to add images in the manuscript. I have few other suggestions as follows -
1. The author must cite the following journal in the section 3.1. http://dx.doi.org/10.1038/354056a0
2. It would be better if the authors add their review the following articles and add their thoughts in the section 3.2.
https://doi.org/10.1016/j.bioactmat.2018.03.001
https://doi.org/10.1177/0021998314534704
3. The author should also discuss the effect of toxicity with respect to the following articles.
DOI: 10.1039/C9TB02861Ghtt
Author Response
Reviewer No. 3
- My first recommendation would be to add images in the manuscript.
Answer: The images have been added.
- The author must cite the following journal in the section 3.1. http://dx.doi.org/10.1038/354056a0
Answer: It has been done (Iijima, S. (1991) Synthesis of Carbon Nanotubes. Nature, 354, 56-58).
- It would be better if the authors add their review the following articles and add their thoughts in the section 3.2.
https://doi.org/10.1016/j.bioactmat.2018.03.001 Sharmeene
https://doi.org/10.1177/0021998314534704 Salem
Answer: it has been done.
- The author should also discuss the effect of toxicity with respect to the following articles.
DOI: 10.1039/C9TB02861Ghtt
Answer: Regretfully, we have found such a DOI in neither Web of Science nor the Internet (Google). However, we think that we have deeply discussed the subject of the possible toxicity of CNTs, still questioned or not.
All changes have been shown in the revised manuscript (version “Detect changes”). Besides, we have put to the resubmitted manuscript also the new version after the recommendation “Accept all changes”.
Round 2
Reviewer 1 Report
The quality of this review paper has improved a lot after revision. The corrections and comments are correct and satisfying. This paper is publishable in this corrected form.
Author Response
Thank you
Reviewer 2 Report
Authors's reply is acceptable and it is publishable now.
Author Response
Thank you